# Evaluation of Household Preparedness and Risk Factors for Cutaneous Leishmaniasis (CL) Using the Community Assessment for Public Health Emergency Response (CASPER) Method in Pakistan

**DOI:** 10.3390/ijerph19095068

**Published:** 2022-04-21

**Authors:** Muhammad Numan, Shumaila Naz, Rehama Gilani, Azhar Minhas, Haroon Ahmed, Jianping Cao

**Affiliations:** 1Department of Biosciences, COMSATS University Islamabad (CUI), Park Road, Chak Shahzad, Islamabad 45550, Pakistan; nomi.malik18@yahoo.com; 2Department of Biological Sciences, National University of Medical Sciences (NUMS), Rawalpindi 46000, Pakistan; shumaila.naz@numspak.edu.pk (S.N.); rehma.gilani@numspak.edu.pk (R.G.); 3Department of Dermatology, Combined Military Hospital (CMH), Quetta 87300, Pakistan; azharminhasmd@gmail.com; 4National Institute of Parasitic Diseases, Chinese Center for Disease Control and Prevention (Chinese Center for Tropical Diseases Research), Shanghai 200025, China; 5Key Laboratory of Parasite and Vector Biology, National Health Commission of the People’s Republic of China, Shanghai 200025, China; 6World Health Organization Collaborating Center for Tropical Diseases, Shanghai 200025, China; 7The School of Global Health, Chinese Center for Tropical Diseases Research, Shanghai Jiao Tong University School of Medicine, Shanghai 200025, China

**Keywords:** CASPER, cutaneous leishmaniasis, household, sandfly, risk factors

## Abstract

(1) Background: In endemic areas of Pakistan, local community knowledge and attitudes towards cutaneous leishmaniasis (CL) are critical elements in the effective control and management of the disease. A cross-sectional epidemiologic design was used to assess the disease concern, preparedness, practices, and preventive behavior of the households and to assist the personnel and health care professionals in strengthening their planning efforts and awareness of CL. (2) Methods: A two-stage cluster sampling process, i.e., Community Assessment for Public Health Emergency Response (CASPER) was conducted from September 2020 to March 2021 on present household-level information about community needs and health status regarding CL in a cost-effective, timely, and representative manner. (3) Results: In the current study, 67% of the respondents were aware of CL and its causative agent and showed a low level of pandemic preparedness. The majority (74%) of the respondents mentioned that they did not avoid sandfly exposure areas. The majority (84%) of respondents had unsatisfactory behavior towards using bed nets, sprays, or repellents. (4) Conclusion: In endemic areas of Pakistan, the inadequate concern and low preparedness of the local community toward CL are critical aspects in efficient control and management of the disease.

## 1. Introduction

Leishmaniasis is a neglected tropical disease caused by protozoan trypanosomatidae parasites of the genus Leishmania, which are transmitted naturally through the bite of infected female sand flies [1]. There are more than 20 disease-causing species of Leishmania [2,3] that have different clinical forms including cutaneous leishmaniasis (CL), mucocutaneous leishmaniasis (MCL), diffuse cutaneous leishmaniasis (DCL), and visceral leishmaniasis (VL, also known as kala-azar) [4,5]. Leishmaniasis has been documented in 98 countries, affecting approximately 350 million people globally, making it one of the world’s most frequent and significant neglected tropical diseases [6,7]. CL has a wide range of distribution and presents between 0.7 and 1.2 million cases reported worldwide each year [8]. More than 1.5 million leishmania cases are reported annually with 0.7–1.2 million cases of CL occurring in Afghanistan, Algeria, Iran, Tunisia, Brazil, Pakistan, Iran, and Saudi Arabia [9]. A high rate of poverty, a large population of immigrants, geographic proximity to the endemic regions, and a favorable climate for sandfly life cycle are all crucial variables involved in leishmania transmission [10]. 

CL is growing rapidly and is one of the most serious public health concerns in Pakistan. CL is extremely endemic in remote areas of Balochistan, Punjab, Sindh, and throughout Khyber Pakhtunkhwa (KPK). It is especially endemic in the Lower and Upper Dir districts, the regions bordering Afghanistan, and localities with the highest number of Afghan refugees [11]. The health authorities of Pakistan have reported 21,000 to 35,000 cases of both anthroponotic (ACL) and zoonotic (ZCL) variants of CL [8,12]. There have been various reports of atypical manifestations of CL disease either owing to unusual lesion sites or their distinct morphology [13]. In Pakistan, information on risk factors in various endemic areas are scarce. There is lack of awareness regarding the sandfly vector and the possibility of scarring despite therapy. The search for traditional therapies for CL treatment and non-reporting to healthcare providers are also included in the risk factors. [14]. Therefore, determining the prevalence of risk variables is one approach to alert public health and healthcare officials to the potential impact of CL in their areas. Although passive surveillance and monitoring of vector population are assisting and guiding policy decision makers in determining priorities, these interventions do not provide a complete picture of Pakistan’s current CL disease impact. Therefore, a systemic data collection of the risk factor prevalence, community preparedness, and practices is required to determine the current and potential future burden of leishmaniasis [15]. 

To achieve these goals, the Community Assessment for Public Health Emergency Response (CASPER) technique was used. CASPER is an epidemiological practice that provides information on preparedness, underlying vulnerabilities, and community attitudes at the household level [16]. Assessing the community’s level of preparedness can benefit public health because data gathered can be utilized for financial support and future planning. In response to the potential threat posed by CL, the most significant requirement is the participation of the community in control programs and successful prevention [17]. Therefore, the current study aimed to validate the efficacy of CASPER by assessing the preparedness and prevalence of CL risk factors in Layyah District, Dera Ghazi Khan, Punjab where the epidemic of CL has been reported [18,19] but no exact data on disease endemicity are available. To date, no such study has been conducted to highlight the preparedness and awareness of the population regarding CL and vector despite the endemicity of sandflies in Pakistan. 

The present study aims to determine the disease concern, preparedness, practices, and preventive behavior of the households of the Layyah District, Dera Ghazi Khan, Punjab, Pakistan where no such study has been carried out and to date no information is available. This is the first CASPER study regarding CL in Pakistan and the data will fill the gap. The data can be used to make informed decisions, allocate resources, and address vector elimination strategies for livestock owners and residents with specific demographic/household characteristics. This knowledge could help guide future health education activities aimed at reducing leishmaniasis in Pakistan’s endemic areas.

## 2. Material and Methods

### 2.1. Study Area

The Layyah District, which is part of the Sindh Sagar Doaba, is a semi-rectangular tract of sandy territory located between the Indus and Chenab rivers to the north and south. It is located between the longitudes of 70°44′ and 71°50′ E and the latitudes of 30°45′ and 31°24′ N. Layyah, Karor, and Chaubara are the three tehsils that makeup Layyah, shown in Figure 1. The research area’s northern portion features sandy topography, while the southern portion is inundated due to summer floods. It is highly hot in the summer and extremely chilly in the winter. The four seasons are summer (May to July), autumn (August to October), winter (November to January), and spring (February to April). June is the hottest month, with a maximum temperature of 51 °C, while December is the coldest, with a minimum temperature of 2 °C. The maximum yearly rainfall in the area is 21 mm. At the lowest point, the relative humidity is 33.4 percent and at the highest point it is 66.6 percent. Farmers’ main agricultural ventures are cattle, bullocks, goats, camels, and sheep [20,21]. 

### 2.2. Ethical Approval

The study was approved by IRB & Ethics Committee of National University of Medical Sciences (NUMS), Rawalpindi, Pakistan under reference number 06/R&D/NUMS. Before data collection, the respondents were clearly informed about the objectives of study and verbal consent from each head of household was taken.

### 2.3. Study Design 

A CASPER survey using two-stage cluster sampling was used to highlight the preparedness and awareness of the population of Layyah District, Pakistan regarding CL. The sampling frame was taken from the Layyah District union council. Rural areas were visited during the study duration, and data were collected using questionnaires. To choose a representative sample of families to be questioned, we randomly chose 30 clusters (census blocks) in Tehsil Lal Esan in the first stage with a probability proportional to the number of households within each cluster. In the second stage, systemic random selection was used to select eighteen houses from each of the clusters that had been chosen. The population data are given in Appendix A.

### 2.4. CASPER Survey 

To collect the data, a detailed questionnaire was developed. After all, participants gave their informed consent, the data were collected using the interview approach. Interviewing, data collecting, and record-keeping were all taught to a team. To ensure data collection and record-keeping, the interview procedure was routinely coordinated with the supervisor. The participants were informed about the study’s purpose before the data collection. The data were collected from September 2020 to March 2021. The CASPER survey of CL was primarily focused on disease concern, preparedness, practices, and preventive behavior of households. The survey was composed of 50 questions and included sections related to demographics, household knowledge including vector concerns, physical and behavioral status, travel history, housing characteristics, prevention, pets, and household needs. In our study population, the data collection technique was also evaluated for consistency and validity. A pilot analysis was carried out in selected areas among 30 participants from study the area before the questionnaire was finalized. These participants were asked to judge the questionnaire’s phrasing, appropriateness, and clarity. The structured questionnaire has high internal consistency, with a Cronbach’s alpha value of 0.8. The pretest data were excluded from the final analysis. The data were collected through door-to-door visits and by conducting face-to-face interviews with willing participants.

### 2.5. Participation and Population Size

Generalized populations from different clusters were selected and respondents were also randomly selected from rural areas (locally known as chaks). The WHO sample size calculator was used with the following assumptions: 5% allowed error, 95% confidence interval with a statistical significance (*p*-value) of 0.05 [22]. The sample size for this study was estimated to be 540 and eligible respondents who resided in the selected household for response were aged 18 years or older. Based on exclusion criteria, 2.7% (15) respondents were below the age of 18 years, and 4.6% (26) questionnaires were incomplete. After exclusion, the sample size of the study was reduced to 500 with a 95% confidence interval and a statistical significance (α-value) of 0.05. As per CASPER criteria, the sample size should be 210 but, in our study, the population size is 500 as our population size is greater and the confidence in our estimate increases, uncertainty decreases, and we have greater precision. The sample size before and after exclusion criteria is given in Appendix A. 

### 2.6. Data Analysis

A database was created by entering data into a Microsoft Excel worksheet. SPSS (SPSS Inc., Chicago, IL, USA) version 26.0 was used to analyze the data. The respondents’ gender’s frequency and percentage were reported using the socio-demographic data. The association of the socio-demographic characteristics to disease concern, preparedness, practices, and preventive behavior of the households was analyzed using the chi-squared test. The results were recorded as frequencies and *p*-values. For all purposes, 0.05 *p*-values were considered as a level of significance. 

## 3. Results 

### 3.1. Socio-Demographic Characteristics of the Households

In this survey, a total of 500 respondents were included. Missing data were reported in less than 30 participants. The frequency of male participants (*n* = 403) was higher than female (*n* = 97) and most of the respondents were in the age group of 18–28 and 29–38. Approximately 72.8% of the households lived in a nuclear family system, with a majority of the households having a male as the head (72.6%). More than 50% of the households had one or more children aged younger than two years old, and roughly 64% of the households had a family size of 5–8 total members, 14.6% had 1–2 members in their family and only 20.6% had >8 members. The largest ethnicity group of the respondents was native, i.e., Saraiki, followed by Punjabi and then Pathan. The educational characteristics of the surveyed population revealed that a plurality (30.6%) of respondents were illiterate and just (8.4%) were graduated. Only 39.6% (*n* = 198) of the respondents had a monthly income of around 25,000–50,000 PKR. The socio-demographic characteristics of the households are given in Table 1.

### 3.2. Disease Concerns of Households

Only 43.6% (*n* = 218) of the respondents had heard of the zoonotic disease and 56.4% (*n* = 282) had not heard about the disease. Likewise, 33% (*n* = 165) had knowledge about CL and 67% (*n* = 335) were unaware of it. The majority of the participants did not know that sandflies carry this disease. Only 17.4% (*n* = 87) of the respondents knew the peak time for sandfly biting is dawn and few 29.8% (*n* = 149) of the respondents were aware that camping in the desert increases the risk of disease spread. When the respondents were questioned about the risk factors, 27.6% (*n* = 138) answered poor hygiene, 35.8% (*n* = 179) said sleeping in an open area, 10.2% (*n* = 51) walking barefoot, and 20.2% (*n* = 101) answered all of these. According to 70.4% (*n* = 352) of the respondents, CL is not a curable disease while 29.6% (*n* = 148) had the opinion that CL was a curable disease. Detail regarding the disease concern of households is given in Table 2. 

### 3.3. Preparedness of Households 

Out of 500 respondents, only 37.8% (*n* = 189) of the respondents’ agreed that CL might be treatable with traditional and herbal medicines while 62.2% (*n* = 311) disagreed. Similarly, 36.8% (184) were in favor of the health department taking actions to prevent CL disease and raise public awareness through education. The majority 66.6% (*n* = 333) of the respondents had not heard of these types of surveys. About 7.4% (*n* = 37) of the respondents had known about surveys through social media, 7.8% (*n* = 39) through websites and 18.2% (*n* = 91) heard about it through friends or family. The details are given in Table 3. 

### 3.4. Practices and Environmental Characteristics of Households

A total of 29.6% (*n* = 148) respondents answered that they maintained hygiene at home by proper floor cleaning, whereas 33.6% (*n* = 168) keep their homes dust free and 26% (*n* = 130) used insecticide sprays. A total of 80.6% (*n* = 403) of the respondents did not prefer nets treated with insecticides. Likewise, 92.6% (*n* = 463) of the respondents did not spray their houses and animal shelters with any spray. Moreover, 60.6% (*n* = 219) of the households kept free-range pets whereas 39.4% (*n* = 142) kept their pets tied up. A total of 44.6% (*n* = 223) of respondents preferred natural remedies as repellents, 22% (*n* = 110) considered repellents to be expensive, 17.2% (*n* = 86) declared their unavailability and 15.6% (*n* = 78) experienced rashes or irritated skin due to mosquito repellents. The detail of practices and environmental characteristics of households are given in Table 4.

### 3.5. Preventive Behavior of Households

A total of 80.4% (*n* = 402) of the respondents did not take any actions to protect themselves while 19.6% (*n* = 98) adopted preventive actions toward CL. Overall, most of the respondents did not adopt any preventive actions toward CL because most of them were impassive when they were asked about preventive practices. However, after obtaining knowledge of CL, 24.8% (*n* = 124) said that they will maintain good hygiene, 6.8% (*n* = 34) responded they well bath daily, 6.2% (*n* = 31) responded they will avoid going outdoor during dawn and dusk, 1.6% (*n* = 8) responded to wear full sleeves, 1% (*n* = 5) said they will stay away from animals and 56.5% (*n* = 282) responded that they would take all measures to prevent themselves from CL. The details are given in Table 5. 

### 3.6. Association of Sociodemographic Characteristics with Disease Concern, Preparedness, Practices, and Preventive Behavior of Households

The association between sociodemographic characteristics, disease concern, preparedness, and preventive behavior of households were analyzed as shown in Table 6. A significant association (*p* < 0.05) was observed with gender, education, ethnicity, income, activities followed by children, number of children with respect to disease concern. There was no significant association observed for age (*p* = 0.51), gender of head of household (*p* = 0.85), and family structure (*p* = 0.17). 

There was also a significant association (*p* < 0.05) of age, ethnicity, activities followed by children, education, monthly income with respect to practice. In the case of preventive behavior, there was a positive association (*p* < 0.05) between gender, age, household members, ethnicity, education, and monthly income. However, non-significant associations between the gender of the household head (*p* = 0.56), number of children in the household (*p* = 0.30), family structure (*p* = 0.27), and preventive practices were observed.

## 4. Discussion

CL is becoming more endemic and disseminating in previously infection-free areas of Punjab, raising the probability that disease-free areas near CL endemic areas are also at risk [13]. Due to their proximity to other CL endemic foci, such as federally managed tribal areas of KPK and Afghanistan, the Upper and Lower Dir Districts are high-risk locations for CL infection [14,23]. The current study attempted to determine the community evaluation of CL among the people of Layyah District, Pakistan and is the first community assessment of leishmaniasis to our knowledge. This household data could be used as a starting point for evaluating and implementing CL control strategies in these or comparable areas. Furthermore, the current study is one of the greatest in terms of sample size (total of 500 respondents), with representative samples from across the Layyah.

In the conducted survey, 33% of the respondents were aware of CL, but their level of concern was so low that 79.4% said they did not know enough about the disease’s causative agent. In contrast to our study, in Saudi Arabia general awareness of CL was satisfactory, with 76% of respondents at least knowing of the disease’s infectious nature [24]. The majority of the respondents in the current study were either illiterate or had a low level of education, which could explain their lack of disease concern regarding CL. As an outcome, it must be emphasized that using health educational written material has little benefit in terms of CL prevention; however, using alternative approaches such as interviews is more useful. According to studies conducted in India, Nepal, Bangladesh, and Brazil, the vast majority of responders (98%) have significantly improved their knowledge and preparedness for leishmaniasis [25]. This high awareness and concern for CL in these kinds of areas is likely due to the fact that they are endemic areas with many cases of the disease.

The sandfly has been proven to be anthropophilic, feeding on animals, rats, and poultry in addition to human blood [26]. The majority (65%) of respondents of the surveyed houses disposed of or throw the garbage on the street every day. Waste has been thought to be a breeding ground for sand flies. Reported studies agreed that inadequate dwelling conditions and poor waste management were believed to be suitable for sandfly reproduction [27,28]. When asked about their knowledge of the route of transmission, a large proportion of respondents (74%) did not avoid sandfly-infested locations, whereas 60.6% kept free-range animals. In our study, the low awareness of the sandfly vector as a carrier of CL is consistent with previous research conducted in Al-Ahsaa, Saudi Arabia, where only 37.4% of respondents could identify the sandfly as a carrier of CL. Contrary to our result, in Isfahan knowledge regarding CL was very high, with 97.9% of respondents knowing that sand flies carry the disease [29]. The present study’s uncertainty regarding vector transmission is similar to a study in Nepal’s rural areas, which indicated that most villagers believed that the mosquitoes, not the sand flies, were responsible for illness transmission [30]. 

Amongst the respondents of the present study, a majority (70.2%) used to sleep outside or camping, which are two possible risk factors for encountering CL. On the other hand, a study conducted in the Al-Ahsaa region, 60% of the respondents were well aware of risk factors associated with CL [24]. Furthermore, sleeping outside in the open air raised the risk of human CL, most likely due to sandfly bites while sleeping. Sleeping outside in open space during the summer months (May–September) is prevalent in Pakistan, and many people, particularly in rural areas, choose to do so. Sandfly activity increases in June and July, reaching a peak in August. Furthermore, entomological studies show that sand flies’ nocturnal activity begins early in the evening and is significantly linked to relative humidity rather than temperature [31].

Regarding prevention of CL, the majority (92.6%) of respondents living in houses had no knowledge of suitable preventive measures and they did not prefer to spray in their animal shelters or houses. A total of 85.2% of respondents mentioned that they did not wear protective clothing for prevention whilst 80.4% declared that they did not take any preventive measures to protect themselves from disease. Only 12.4% mentioned that the use of mosquito coil is necessary for the prevention of disease. In the Al-Ahsaa region, 33.1% of the respondents assumed the relevance of protective gear, 13.4% recognized the use of protective netting, and 11.8% recognized the preventive role of insecticide spraying, similar to the current study [24]. Only 11.8% of our respondents thought CL was a major health issue. Although the indoor personal prophylactic measures utilized by respondents in our study were not sufficient, 19.4% of them used insecticide-treated nets when sleeping. This is in line with Hejazi’s study conducted in Isfahan, which discovered unsatisfactory behavior among the community they surveyed when it came to preventive measures like utilizing a bet net or repellents [32]. 

Furthermore, few respondents assumed that CL can be treated medically whereas 62.2% responded that CL is a self-limited disease and 37.8% of them thought CL is can be treated by herbal preparations, which is incorrect [33]. Furthermore, in southern Iran, a significant percentage of the respondents (21%) believed in rational medicine as a treatment for CL [34]. Unfortunately, our study showed that 70.4% of respondents wrongly believed that CL cannot be cured, which concurs with another study in Pakistan that reported that 42% of respondents considered it to be a fatal disease [10]. The communities in underdeveloped nations, such as Pakistan, must be educated and informed about leishmaniasis and its vectors through radio, television, surveys, and educational programs. The low perceived preparedness and unsatisfactory practices of the study population highlight the need for health education, awareness campaigns, and future disease research to design appropriate policies to guide government and stakeholders in reducing the risk of cutaneous leishmaniasis outbreaks in such areas [34,35]. Due to the case-control nature of this study, the findings have certain limitations as it is difficult to prove temporal causation in case-control studies. Furthermore, selection and recollection biases are common in these study designs. To determine the causal association between risk factors and outcome, future studies based on cohort study design would be more appropriate.

## 5. Conclusions

The current study proposed that the population of Layyah has inadequate preparedness as well as a lack of requisite preventive behavior and practices against CL infection, making them prone to disease. The respondents also have limited knowledge related to etiology, risk factors, and treatment facilities for CL. Hence, this study emphasizes the importance of educating this community on preparedness and preventative steps that are necessary to avoid infection, with a focus on vector management. Furthermore, people should be educated through face-to-face education and the use of instructional aides, as written material may not be effective to overpass this gap. CL has developed as a challenging infection in Pakistan, requiring the development of large-scale preventative measures and public awareness to restrict disease spread.

## Figures and Tables

**Figure 1 ijerph-19-05068-f001:**
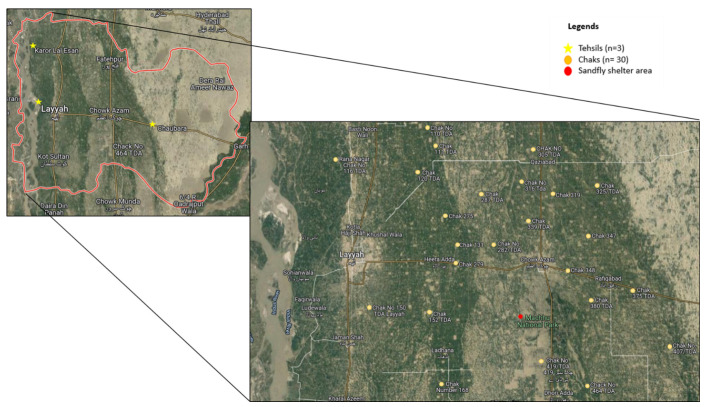
Map of the study area depicting sampling areas where samples were collected.

**Table 1 ijerph-19-05068-t001:** Socio-demographic characteristics of the households.

Sr. No	Socio-Demographic Characteristics	Frequency (*n*)	Percentage (%)
1	Gender Male Female	403 97	80.6 19.4
2	Age 18–28 29–38 ≥39	168 175 157	33.6 35 31.4
3	Household head Male Female	363 137	72.6 27.4
4	Household members 2–4 5–8 >8	73 324 103	14.6 64.8 20.6
5	No of children in the household 1–3 4–6 >6	253 219 28	50.6 43.8 5.6
6	Ethnicity Punjabi Saraiki Pathan	181 251 68	36.2 50.2 13.6
7	Activities followed by children Indoor Outdoor	160 340	32 68
8	Education College/University High School Secondary Primary Illiterate	42 81 77 153 147	8.4 16.2 15.4 30.6 29.4
9	Family structure Nuclear family Extended family	364 136	72.8 27.2
10	Monthly income (PKR) <10,000 10,000–25,000 25,000–50,000 >50,000	45 156 198 101	9 31.2 39.6 20.2

**Table 2 ijerph-19-05068-t002:** Disease concern of the households.

Variables	Frequency (*n*)	Percentage (%)
Disease Concern		
Have you ever heard about Zoonotic disease? Yes No	218 282	43.6 56.4
Do you know about cutaneous leishmaniasis (CL)? Yes No	165 335	33 67
Did you ever see a leishmaniasis patient or have a history of leishmaniasis? Yes No	56 444	11.2 88.8
Do you think cutaneous leishmaniasis is a lethal disease? Yes No	31 469	6.2 93.8
Do you know about sandflies? Yes No	235 265	47 53
Can you identify/differentiate sandflies from common houseflies/mosquitoes? Yes No	210 290	42 58
Do you think sandflies carry this disease? Yes No	103 397	20.6 79.4
What is the peak incidence time? Morning Evening Dawn to dusk Night Do not Know	87 101 170 65 77	17.4 20.2 34 13 15.4
Do you know about the life cycle of sandflies? Yes No	59 441	11.8 88.2
In your opinion, CL is more common among which gender? Male Female Both Don’t Know	19 36 326 119	3.8 7.2 65.2 23.8
Who is the at most at risk from CL? Pregnant women Women of childbearing age 15–44 Disable people. Adolescence 15–24 Children Older Adult Everybody Do not Know	16 2 37 7 89 2 254 93	3.2 0.4 7.4 1.4 17.8 0.4 50.8 18.6
What are the risk factors for cutaneous leishmaniasis? Poor hygiene House architecture Walking barefoot Sleeping in an open area All above Don’t Know	138 12 19 51 179 101	27.6 2.4 3.8 10.2 35.8 20.2
Where do you and members of your household typically get bitten by sandfly/mosquitoes? Home Work School Park Others	282 103 49 58 8	56.4 20.6 9.8 11.6 1.6
Does camping in the desert⁄ farm (in an open area) increase the risk for disease? Yes No	149 351	29.8 70.2
Can this disease be cured? Yes No	148 352	29.6 70.4

**Table 3 ijerph-19-05068-t003:** Preparedness of the households.

Variables	Frequency (*n*)	Percentage (%)
Preparedness		
CL is associated with dusty areas Agree Disagree	202 298	40.4 59.6
CL is a serious health problem in this community Agree Disagree	59 441	11.8 88.2
CL is treated by traditional and herbal preparations Agree Disagree	189 311	37.8 62.2
Does your household have adequate drinking water? Yes No	500 0	100 0
Do you or members of your family hear about this survey prior to us talking to you today? If yes, how did you or your HH member hear about it Social media Website Press release Family/friends No	37 39 0 91 333	7.4 7.8 0 18.2 66.6
What actions do you believe the health department should take to prevent Cutaneous Leishmaniasis disease? Education/Awareness Inspection for waste management Spraying for sandflies	184 173 143	36.8 34.6 28.6

**Table 4 ijerph-19-05068-t004:** Practices and environmental characteristics of the households.

Variables	Frequency (*n*)	Percentage (%)
Practices and Environmental Characteristics		
How your family disposes waste or garbage every day? Throw garbage on the street Throw in bins Burn it properly	325 95 80	65 19 16
How do you maintain good hygiene at home? Proper floor cleaning Spray at home Keeping house dust free All above None	148 5 49 168 130	29.6 0.1 9.8 33.6 26
Do you walk barefoot at home? Yes No	466 34	93.2 6.8
Do you have access to the forest? Yes No	282 218	56.4 43.6
Do you prefer spraying your houses and animal shelters? Yes No	37 463	7.4 92.6
Do you prefer the use of nets treated with insecticides while sleeping? Yes No	97 403	19.4 80.6
From what type of material your house is made up of Wood Bricks Stones Mud	1 399 3 97	0.2 79.8 0.6 19.4
The floor of your house made up of Bricks Sand Stone	115 172 213	23 34.4 42.6
Does the waste present around your house? Yes No	376 124	75.2 24.8
Do you avoid areas of mosquito exposure? Yes No	130 370	26 74
Do you have pets? Yes No	361 139	72.2 27.8
How do you keep your pets? Free range Tied	219 142	60.6 39.4
Do you use skin repellent? Yes No	80 420	16 84
What, if any, are barriers to using mosquito repellent? Too expensive Product not available Prefer natural remedies Rashes or irritated skin All above	110 86 223 78 3	22 17.2 44.6 15.6 0.6
What is your HH current source of drinking water? Unfiltered tap Filtered tap water	430 70	86 14

**Table 5 ijerph-19-05068-t005:** Preventive behavior of the households.

Variables	Frequency (*n*)	Percentage (%)
Preventive Behavior		
Do you take any actions to protect yourselves? Yes No	98 402	19.6 80.4
Do you wear protective clothing? Yes No	74 426	14.8 85.2
Do you avoid being outside at peak times? Yes No	45 455	0.9 91
Do you use burn mosquito coils or candles? Yes No	62 438	12.4 87.6
After knowing this disease, what preventive measures will you take to prevent yourself from this disease? Bath daily Maintain good hygiene Stay away from animals Wear full sleeves Avoiding going outdoor during dusk and dawn All above None	34 124 5 8 31 282 16	6.8 24.8 0.1 1.6 6.2 56.4 3.2

**Table 6 ijerph-19-05068-t006:** Comparison of disease concerns, preparedness, practices, and preventive behavior with socio-demographic characteristics of households.

Socio-Demographic Variables	Disease Concern		*p*-Value
	Adequate *n* (%)	Poor *n* (%)	
Gender Male Female	89 (22.1) 314 (77.9)	43 (44.3) 54 (55.7)	*p* < 0.05
Age 18–28 29–38 ≥39	49 (29.2) 46 (26.3) 37 (23.6)	119 (70.8) 129 (73.7) 120 (76.4)	0.51
Household head Male Female	95 (26.2) 37 (27)	268 (73.8) 100 (73)	0.85
Household members 2–4 5–8 >8	39 (53.4) 79 (24.4) 14 (13.6)	34 (46.6) 245 (75.6) 89 (86.4)	*p* < 0.05
No of children in the household 1–3 4–6 >6	93 (36.8) 38 (17.4) 1 (3.6)	160 (63.2) 181 (82.6) 27 (96.4)	*p* < 0.05
Ethnicity Punjabi Saraiki Pathan	51 (28.2) 77 (30.7) 4 (5.9)	130 (71.8) 174 (69.3) 64 (94.1)	*p* < 0.05
Activities followed by children Indoor Outdoor	74 (46.3) 58 (17.1)	86 (53.8) 282 (82.9)	*p* < 0.05
Education College/University High School Secondary Primary Illiterate	80 (54.4) 35 (22.9) 7 (9.1) 6 (7.4) 4 (9.5)	67 (45.6) 118 (77.1) 70 (90.9) 75 (92.6) 38 (90.5)	*p* < 0.05
Family structure Nuclear family Extended family	102 (28) 30 (22.1)	262 (72) 106 (77.9)	0.17
Monthly income <10,000 10,000–25,000 25,000–50,000 >50,000	0 (0) 22 (14.1) 76 (38.4) 34 (33.7)	45 (100) 134 (85.9) 122 (61.6) 67 (66.3)	*p* < 0.05
**Preparedness**			
	**Adequate *n* (%)**	**Poor *n* (%)**	
Gender Male Female	77 (19.1) 38 (39.2)	326 (80.9) 59 (60.8)	*p* < 0.05
Household head Male Female	88 (24.2) 27 (19.7)	275 (75.8) 110 (80.3)	0.28
Household members 2–4 5–8 >8	39 (53.4) 65 (20.1) 11 (10.7)	34 (46.6) 259 (79.9) 92 (89.3)	*p* < 0.05
No of children in household 1–3 4–6 >6	76 (30) 36 (16.4) 3 (10.7)	177 (70) 183 (83.6) 25 (89.3)	*p* < 0.05
Ethnicity Punjabi Saraiki Pathan	53 (29.3) 55 (21.9) 7 (10.3)	128 (70.7) 196 (78.1) 61 (89.7)	*p* < 0.05
Activities followed by children Indoor Outdoor	56 (35) 59 (17.4)	104 (65) 281 (82.6)	*p* < 0.05
Education College/University High School Secondary Primary Illiterate	64 (43.5) 29 (19) 9 (11.7) 9 (11.1) 4 (9.5)	83 (56.5) 124 (81) 68 (88.3) 72 (88.9) 38 (90.5)	*p* < 0.05
Family structure Nuclear family Extended family	98 (26.9) 17 (12.5)	266 (73.1) 119 (87.5)	*p* < 0.05
Monthly income (PKR) <10,000 10,000–25,000 25,000–50,000 >50,000	2 (4.4) 21 (13.5) 64 (32.3) 28 (27.7)	43 (95.6) 135 (86.5) 134 (67.7) 73 (72.3)	*p* < 0.05
**Practices**			
	**Adequate *n* (%)**	**Poor *n* (%)**	
Gender Male Female	40 (9.9) 12 (12.4)	363 (90.1) 85 (87.6)	0.47
Age 18–28 29–38 ≥39	23 (13.7) 23 (13.1) 6 (3.8)	145 (86.3) 152 (86.9) 151 (96.2)	*p* < 0.05
Household head Male Female	39 (10.7) 13 (9.5)	324 (89.3) 124 (90.5)	0.68
Household members 2–4 5–8 >8	13 (17.8) 31 (9.6) 8 (7.8)	60 (82.2) 293 (90.4) 95 (92.2)	0.07
No. of children in household 1–3 4–6 >6	29 (11.5) 20 (9.1) 3 (10.7)	224 (88.5) 199 (90.9) 25 (89.3)	0.70
Ethnicity Punjabi Saraiki Pathan	26 (14.4) 24 (9.6) 2 (2.9)	155 (85.6) 227 (90.4) 66 (97.1)	*p* < 0.05
Activities followed by children Indoor Outdoor	27 (16.9) 25 (7.4)	133 (83.1) 315 (92.6)	*p* < 0.05
Education College/University High School Secondary Primary Illiterate	35 (23.8) 13 (8.5) 2 (2.6) 2 (2.5) 0 (0)	112 (76.2) 140 (91.5) 75 (97.4) 79 (97.5) 42 (100)	*p* < 0.05
Family structure Nuclear family Extended family	35 (9.6) 17 (12.5)	329 (90.4) 119 (87.5)	0.34
Monthly income <10,000 10,000–25,000 25,000–50,000 >50,000	0 (0) 3 (1.9) 31 (15.7) 18 (17.8)	45 (100) 153 (98.1) 167 (84.3) 83 (82.2)	*p* < 0.05
**Preventive behavior**			
	**Adequate *n* (%)**	**Poor *n* (%)**	
Gender Male Female	55 (13.6) 22 (22.7)	348 (86.4) 75 (77.3)	*p* < 0.05
Age 18–28 29–38 ≥39	41 (24.4) 25 (14.3) 11 (7)	127 (75.6) 150 (85.7) 146 (93)	*p* < 0.05
Household head Male Female	58 (16) 19 (13.9)	305 (84) 118 (86.1)	0.56
Household members 2–4 5–8 >8	19 (26) 54 (16.7) 4 (3.9)	54 (74) 270 (83.3) 99 (96.1)	*p* < 0.05
No of children in household 1–3 4–6 >6	45 (17.8) 29 (13.2) 3 (10.7)	208 (82.2) 190 (86.8) 25 (89.3)	0.30
Ethnicity Punjabi Saraiki Pathan	33 (18.2) 43 (17.1) 1 (1.5)	148 (81.8) 208 (82.9) 67/98.5	*p* < 0.05
Activities followed by children Indoor Outdoor	32 (20) 45 (13.2)	128 (80) 295 (86.8)	*p* < 0.05
Education College/University High School Secondary Primary Illiterate	48 (32.7) 19 (12.4) 6 (7.8) 4 (4.9) 0 (0)	99 (67.3) 134 (87.6) 71 (92.2) 77 (95.1) 42 (100)	*p* < 0.05
Family structure Nuclear family Extended family	60 (16.5) 17 (12.5)	304 (83.5) 119 (87.5)	0.27
Monthly income <10,000 10,000–25,000 25,000–50,000 >50,000	0 (0) 12 (7.7) 42 (21.2) 23 (22.8)	45 (100) 144 (92.3) 156 (78.8) 78 (77.2)	*p* < 0.05

## Data Availability

The data can be requested from the authors.

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
