# Peer review of "Evaluation of Household Preparedness and Risk Factors for Cutaneous Leishmaniasis (CL) Using the Community Assessment for Public Health Emergency Response (CASPER) Method in Pakistan"

_ijerph, 2022, doi:10.3390/ijerph19095068_

Round 1
Reviewer 1 Report
Numan et al. provide an epidemiological study targeting people at risk of cutaneous leishmaniasis in a specific region in Pakistan. The CASPER approach and the writing style (clear and concise) are manuscript’s strengths. However, I’ve noted some issues, minor and major, in the process of review.
Major
1, authors should provide to the readers the actual prevalence of cutaneous leishmaniasis in the targeted population. Did the authors collect clinical data from this population?
2, it is not clear whether the utilized instrument (questionnaire) for obtaining the data of this manuscript has been previously validated to the targeted population. Otherwise, results can be flawed.
Minor
1, Figure 1 is not acceptable. Please include important geographical elements, including forested regions of the districts that serve as shelters to sandflies.
2, study design using the 2-stage cluster approach is not clear. Please illustrate the sampling approach using Figure 1.
3, Please include sample size equation used for calculation.
4, supplementary tables were not provided.
5, please improve table formatting. See Table 3.
Reviewer 2 Report
The manuscript entitled “Evaluation of household preparedness and risk factors for cutaneous leishmaniasis (CL) using community assessment for public health emergency response (CASPER) method in Pakistan” by Numan et al, describes the use of a 2-stage cluster sampling process CASPER, which was conducted from September 2020 to March 2021 to present household-level information about community needs and health status regarding CL. The study overall showed a high level of disease awareness but a low level of disease preparedness ie avoidance of sandfly exposure areas, un-satisfactory behaviors towards using the bed nets, sprays, or repellents.
Overall, the manuscript is informative and well written. It is of interest and addresses a future scope needed to minimize the spreading of CL in Pakistan and this is better education. Some minor spelling, grammatical mistakes and typos should be corrected.
Reviewer 3 Report
1. Summary
The paper investigates, using the CASPER framework, the concern, preparedness, practices and preventive behaviour of households towards cutaneous leishmaniasis in endemic ares of Pakistan. From Sept 2020 to Mar 2021, households were randomly selected from 30 clusters for collecting household-level information. A low level of pandemic preparedness, none avoidance of sandfly exposure, unsatisfactory usage of bed nets, sprays, or repellents across all households summarise the results obtained. The given conditions make an efficient control and management of CL difficult.
The paper highlights the gap in evidence and develops arguments for using the CASPER framework for data collection and reporting. Methods are fully and clearly described, with the exception of the sample size paragraph. Analysis methods could be more detailed (for instance name of correlation test used should be given). Authors may work on better summarising and reporting the results, and avoid redundancies in tables and main text. Tables should be smaller, if possible. Discussion part is well done, however some research on knowledge, attitudes, and practices have been conducted in Pakistan earlier.
A Cross-Sectional Survey of Knowledge, Attitude and Practices Related to Cutaneous Leishmaniasis and Sand Flies in Punjab, Pakistan
Akram A, Khan HAA, Qadir A, Sabir AM (2015) A Cross-Sectional Survey of Knowledge, Attitude and Practices Related to Cutaneous Leishmaniasis and Sand Flies in Punjab, Pakistan. PLOS ONE 10(6): e0130929. https://doi.org/10.1371/journal.pone.0130929
2. Figures, tables, and graphs
Figure 1:
useful, but impossible to read in the pdf version
Table 1:
Missing answers not reported?
“Family structure” to be defined, explained
“monthly income” currency should not miss.
Either report the numbers in the tables or report the outcome of the analysis in the plain text. Doing both is redundant.
Table 2:
Either report the numbers in the tables or report the analysis in the plain text. Doing both is redundant.
Consider to add your table 2 in supplementary files. Size of table is too large for plain text section.
Table 3:
Please indicate where to read frequencies (n=) or percentages (%) in the table or in the legend.
Again consider to add your table in supplementary files; the information is helpful but exhaustive. Be careful not to be redundant with the main text. The column titles are not formatted adequately (adequate and poor often change place).
Supplementary table 1 no comment
Supplementary table 2 careful with consistent number formats "1,000 or 1000"
3. Specific paragraphs
Results are given in tables and in the main text. However, results in the main text mainly repeat outcomes given in the tables. The authors should revise this redundancy, either by adding the tables in the supplementary files sections or summarising other results in the tables.
Line 116 “this time” do you mean “the study duration”?
Line 141 missing literature reference for “WHO calculator”
Line 143 “50% of outcome factor (p)” - please detail.
Line 147ff unclear statement. Why the sample size 210?
Line 158 Please detail the name of the correlation test used.
4. Language
Line 53 to 56 Language: review the structure of the sentence and consider making two or three sentences out of one (6 times “and”).
Line 58: Language: … which estimated 21,000 to 35,000 “cases of” both ACL and ZCL variants of the disease.
Line 60: Language: … "In Pakistan, information on risk factors in various endemic areas are scarce ..."
Line 60 to 64: Language: review the structure of the sentence and consider making two or three sentences out of one.
Line 64: use “is” instead of “could be”
Line 66: Language: "Although passive surveillance and monitoring of vector population are assisting and guiding policy decision makers in determining priorities, these interventions do not provide a complete picture of Pakistan’s current CL disease impact."
Line 72: Language: “was used [in the present study]” instead of “can be used”.
Line 73: Language: …. "that provides information on preparedness, underlying" ….
Line 75: Language:…."can benefit public health, “because" data gathered" …
Line 86 to 89 Language: review the structure of the sentence and consider making two or three sentences out of one (4 times “and”).
Line 153: Language: reformulate the sentence. What is the “perception of frequency”… for example. Did you mean: "The respondents’ gender’s frequency and percentage was reported using the socio-demographic data."
Line 199 Language
Line 246: Language: do not use the “no.” abbreviation in the main text.
Line 254: Language: “probability” instead of “possibility”.
Line 275: Language: Revise sentence.
Round 2
Reviewer 1 Report
The authors have made improvements on their manuscript. Now only minor revisions are left uncorrected.
In the first response to my question, it is said no exact prevalence is known, but they cite two references. Could the authors bring any clinical data evidence from these references to their own manuscript? What do these references have in terms of clinical data? Ref. 19’s title “Use of mental imagery in psychotherapy: A critical review” seems misplaced, it is in the right place? What does it have on clinical data?
About data validation. It is shown that the questionnaire has been previously validated. Results of analysis of questionnaire consistency should be shown. What is the Cronbach’s alpha of the questionnaire in the validation step?
Sample size calculation is still unspecified. How has the sample size of 540 been calculated? What is the statistical power of a sample size of 500?
